# Integrating Women and Girls’ Nutrition Services into Health Systems in Low- and Middle-Income Countries: A Systematic Review

**DOI:** 10.3390/nu14214488

**Published:** 2022-10-25

**Authors:** Rachael Menezes, Natasha Lelijveld, Stephanie V. Wrottesley, Eilise Brennan, Emily Mates, Philip T. James

**Affiliations:** Emergency Nutrition Network, Kidlington OX5 2DN, UK

**Keywords:** integration, women’s nutrition, reproductive age, interventions, public health systems

## Abstract

Women’s nutrition has been highlighted as a global priority to ensure the health and well-being of both them and future generations. This systematic review summarises the available literature on the integration of nutrition services for girls and women of reproductive age (GWRA) into existing public health systems across low- and middle-income countries, as well as any barriers to integration. We searched PubMed and Cochrane Database of Systematic Reviews for articles published since 2011 according to eligibility criteria. A total of 69 articles were included. Evidence suggested that several services for GWRA are well integrated into public health systems, including antenatal care services, nutrition education and counselling, and micronutrient supplementation programmes. However, there was limited evidence on the integration of family planning, adolescent health, and reproductive health services. Barriers to integration fell into five main themes: lack of training and capacity building, poor multisectoral linkages and coordination, weak advocacy, lack of M&E systems, and inequity. We identified a lack of evidence and services for non-pregnant GWRA and for women postpartum. Addressing barriers to integration and gaps in nutrition services for GWRA would increase service coverage and contribute to improving health outcomes for GWRA and future generations.

## 1. Introduction

Women’s nutrition has been highlighted as a global priority to ensure the health and well-being of both them, their children and future generations. A maternal diet that meets optimal macro- and micronutrient requirements is associated with improved maternal and child health, as well as reduced maternal and child mortality rates [1]. Furthermore, nutrition of women at all life stages directly impacts the health outcomes of future generations [2]. While the focus of many nutrition services is on pregnant women and mothers, this should be expanded to encompass all girls and women of reproductive age (GWRA) (15–49 years), including adolescent girls and women in the postnatal period, for maximum impact [3].

Current coverage of stand-alone interventions targeting GWRA across many low- and middle-income countries (LMICs) is sub-optimal [4,5]. Common interventions targeting nutrition of GWRA include micronutrient/food supplementation programmes, nutrition education and counselling, and food fortification [3,6]. These evidence-based strategies could significantly contribute to improved nutrition for GWRA if delivered efficiently at scale. Integrating nutrition services for GWRA into existing health systems is likely to be an effective strategy for achieving greater coverage [7]. Existing public health systems accommodate a number of platforms through which women’s nutrition could be integrated and delivered (including antenatal/postnatal clinics, family planning platforms, adolescent health and reproductive health platforms). Scaling-up and integrating women’s nutrition services into delivery platforms such as these would increase the uptake and efficiency of these services, decentralising them and making them more accessible to GWRA.

While integration of other key services into health systems has previously been researched, such as prevention of mother-to-child transmission (PMTCT), HIV and reproductive health services, and community-based management of acute malnutrition (CMAM) services, to the best of our knowledge there are currently no systematic reviews discussing the extent to which nutrition services for GWRA have been integrated into existing health systems across LMICs. Evidence from other sectors, such as PMTCT and CMAM, shows that integration is not only possible but extremely successful, with studies suggesting that accessibility to services can increase drastically post-integration, particularly when paired with supporting community outreach initiatives [8].

### 1.1. Objectives

This review aims to summarise the existing research on the integration of nutrition services for GWRA into health systems and to highlight any barriers to integration.

### 1.2. Key Messages

Current evidence suggests that a number of nutrition services for GWRA are well integrated (including antenatal care services, nutrition education and counselling, and micronutrient supplementation programmes) into public health systems. However, there is minimal peer-reviewed evidence surrounding integration of family planning, adolescent health, and reproductive health services.

Several barriers to integration were highlighted, particularly around inadequate support and poor mechanisms for large-scale supervision of community health workers, insufficient monitoring and evaluation systems, and inequalities in access to services.

Addressing barriers to integration and gaps in nutrition services as well as investing in services for non-pregnant GWRA and women postpartum would increase coverage and contribute to improving health outcomes for GWRA.

## 2. Materials and Methods

### 2.1. Search Strategy

We conducted a systematic search of the peer-reviewed literature in PubMed and the Cochrane Database of Systematic Reviews. Due to budgetary limitations, we restricted our searches to these open access databases. The search terms are detailed in Appendix A and were guided by the “Population, Interventions, Control and Outcome” (PICO) framework presented in Appendix A. We published our full search strategy in the PROSPERO database (CRD42021253003).

### 2.2. Inclusion and Exclusion Criteria

#### 2.2.1. Inclusion Criteria

Peer-reviewed publications published between 2011 and present day.

Studies describing or assessing the integration of both direct and indirect nutrition services into existing health services.

Studies targeting GWRA (defined as 15–49 years old by the World Health Organization (WHO)).

Studies conducted in LMICs.

No restrictions on study design.

No restrictions on language.

#### 2.2.2. Exclusion Criteria

Studies that evaluate stand-alone nutrition interventions that have not been integrated into health systems.

Studies concerning girls <15 years old or women >49 years old.

Grey literature.

Conference abstracts.

### 2.3. Outcomes

We were deliberately agnostic about pre-specified outcomes; rather than assessing the effectiveness of nutrition services for GWRA the main focus was on the process of integrating nutrition interventions into health systems, and we therefore did not expect all relevant literature to include outcome descriptions. As a secondary objective, we synthesised any barriers to integration, including any factors identified by authors that restricted integration of nutrition interventions into health systems (such as lack of adequate training for community healthcare workers, poor monitoring and evaluation systems, etc.). We did not specify measures of effect, however, we did report any documented outcome information from the included studies in our results section.

### 2.4. Screening Process and Selection

The lead author imported the search results into Mendeley reference management software. After de-duplication, the results were screened independently by two reviewers based on their title and abstract. After reconciliation, the lead author screened the remaining results by full text and any relevant information that met the inclusion criteria was extracted (Appendix A) [9].

### 2.5. Data Synthesis

This review was conducted according to Synthesis Without Meta analysis (SWiM) guidelines for systematic reviews [10] and PRISMA guidelines were used as a checklist to guide our reporting (see Appendix A). Main outcomes were discussed in narrative form by (a) intervention type and (b) delivery platform. Any barriers to integration specified by authors were discussed as a secondary outcomes. Outcomes and barriers were reported as they were presented by the authors. We used the MeSH terms within our search strategy to guide the structure of the narrative results section. Due to the expected heterogeneity of eligible literature and outcomes, we did not perform a formal risk of bias assessment for each included piece of literature.

The following data was extracted from all relevant research articles: author, date, country, year of data collection, target population (e.g., age range, urban/rural, pregnant/not-pregnant), intervention (e.g., type of intervention, duration of intervention, specific features), integration (e.g., any details on the integration into existing health systems, barriers), outcomes of interest, classification criteria used and key findings (e.g., author conclusions, limitations/strengths, monitoring and evaluation (M&E) strategies, recommendations).

## 3. Results

The database search was conducted on the 13 May 2021 and produced 3542 results. After duplicates were removed, 3519 results were double screened by title and abstract for relevance; at this stage a total of 3268 results were excluded. Following screening of the remaining 251 full texts, a total of 69 studies were included in this review (Figure 1).

A summary of the literature findings by nutrition intervention is presented in Table 1. A full data extraction table with all included studies is detailed in Appendix A, with a complete summary of findings by delivery platform and nutrition intervention in Appendix A.

We found literature describing integration of five main types of nutrition interventions for GWRA into health services (see Table 1): energy and protein supplementation, food fortification, nutrition education/counselling, home food distribution and micronutrient supplementation (defined as a supplement containing one or more micronutrients, e.g., vitamin A, iodine, iron-folic acid, etc.). Three articles discussed energy and protein supplementation programmes, including a global systematic review of interventions [17], as well as research from the Asia and the Pacific region, that were delivered through government Integrated Child Development Services and primary healthcare systems [4,18]. Studies noted that integration could be facilitated by integrating nutrition indicators and guidelines into national health policies. We identified five articles discussing fortification programmes, with research coming from Asia and sub-Saharan Africa. Specifically, we found that interventions involving iron, folic acid and vitamin A fortification were supported by state level policies and integrated into the government Targeted Public Distribution System, through which foods are distributed to a targeted population below the poverty line [12,14]. A total of 18 articles focused on nutrition education and counselling programmes, the majority of which came from South Asia. These articles also highlighted the need for strengthening policy integration, although many of these programmes had been integrated into Maternal, Newborn and Child Health services, and antenatal care service (ANC) services [18,27,32]. A further 41 articles were included that reported on micronutrient supplementation programmes, and one article reported on home food distribution programmes. The literature reported examples of effective integration into ANC services and public health facilities: community health workers (CHWs) were the workforce through which mainly iron-folic acid (IFA) supplementation programmes targeting GWRA were delivered [41,43]. Overall, we found seven articles discussing CHW programmes, though they were also discussed in other articles alongside different intervention types and delivery platforms.

Several health service delivery platforms were identified through which GWRA could be targeted and could access nutrition services. Only four articles were identified that discussed adolescent health services, however, the authors highlighted that this delivery platform played a vital role in reaching girls and young women, with the literature focusing on anaemia prevention via IFA supplementation and coming from India [18,76] and Indonesia [68,71]. Research regarding integration of nutrition services into reproductive health services was also limited (two articles) and the focus was again on IFA supplementation [72,76]. In the delivery of nutrition programmes ANC services were essential, with 29 articles involving ANC integration being included in this review. This service was generally well integrated into government hospitals and health facilities in a number of countries. Nutrition counselling/education and micronutrient supplementation were the two programmes most commonly integrated into ANC services, which were delivered as part of government health facilities, state hospitals or health clinics [11,19,40]. Five articles were identified that explored integration of women’s nutrition into family planning services. These services were also made accessible to hard-to-reach communities through partnering with CHWs [18].

### Barriers to Integration

Barriers to integration fell under five main themes: lack of training and capacity building, poor multisectoral linkages and coordination, weak advocacy, lack of M&E systems, and inequity. Lack of training and support for CHWs, as well as inadequate supervision mechanisms at scale were identified as barriers by several authors [11,19,40,69,77]. Lack of coordination amongst Ministry of Health directorates limited integration across health systems [69,77], with poor linkages and coordination between programmes and policies, as well as a general lack of health and nutrition policies, acting as a barrier to integration of ANC services and micronutrient supplementation in particular [58]. The unification of multi-sectoral programmes and policies within a country is important in the successful scale-up of essential nutrition services for GWRA. However, authors further highlighted that weak advocacy for nutrition amongst governmental actors and a lack of awareness around the importance of women’s and girls’ nutrition was a barrier, and meant that the most vulnerable GWRA either did not have access to, or chose not to utilise, nutrition services [12,32].

Insufficient M&E systems were a further barrier to integration of nutrition services for GWRA. Lack of efficient monitoring and evaluation of nutrition interventions has restricted abilities to make evidence-based adaptations or improvements to programmes, with negative impacts on the quality of services [74]. Prioritising the updating of information regarding nutrition strategies and indicators in national databases to facilitate data validation and comparison would help inform well-designed nutrition policies and programmes [14]. A final barrier commonly discussed was a lack of attention to equity of intervention coverage, leading to inequity in access to services, as many marginalised GWRA were not catered for [15,74,78,79].

## 4. Discussion

This review found 69 peer-reviewed articles discussing the integration of nutrition interventions for GWRA into public health systems. The nutrition services were included within the following categories: energy and protein supplementation, food fortification, nutrition education and counselling, micronutrient supplementation and home food distribution. Common delivery platforms were via CHW programmes, adolescent health services, antenatal/postnatal care services, family planning services and reproductive health services.

There are several sets of international guidelines that include information on nutrition interventions for women and girls, and which state the importance of their integration within existing public health systems, including the WHO 2016 ANC guidelines [80], the 2022 WHO post-natal care guidelines [81] and the WHO 2019 Essential Nutrition Actions [82], amongst others. However, the existence of international guidelines does not necessarily mean that these are translated into national policies, nor from there into service delivery [83]. In the past, however, there have been many efforts to scale-up and integrate nutrition and health services into public health systems, such as the integration of PMTCT, HIV and reproductive health services and CMAM services which demonstrate that integration can be extremely successful. The published literature further suggests that accessibility to services can dramatically increase post-integration, particularly when paired with community outreach initiatives [8]. Other research has indicated that the success and sustainability of integration hinges on a broad whole-systems approach [84] and must include careful planning of programmes to consider context-specific factors [69]. Further barriers to integration were identified, such as poor M&E systems, lack of training and shortage of healthcare staff, inadequate mechanisms for large-scale supervision and support of community health workers (CHWs), and lack of coordination amongst ministry of health directorates [69,77]. Better understanding of prior attempts to integrate health and nutrition services into existing health systems can allow us to learn valuable lessons as well as highlight the opportunities and potential success of women’s nutrition service integration.

Our review found that the research surrounding integration of energy and protein supplementation into health services is very limited. Although energy and protein supplementation is one of the *Lancet* 2021 series on Maternal and Child Undernutrition’s core intervention strategies, the series acknowledges that evidence of its implementation is moderate and questions remain about how to deliver this intervention to underweight GWRA [6,17,85]. Of the three articles included here that described the integration of energy and protein supplementation, all advocated for the expansion of nutrition policies and implementation of targeted interventions to improve health outcomes in adolescent girls [86]. Additionally, the research from India highlighted that the lack of focus on non-pregnant, non-lactating GWRA was hindering efforts to improve women’s nutrition across the country [18]. Globally, there is a need for pre-conception nutrition interventions to be supported by political will in order to be effective in achieving healthy growth trajectories in future generations [87].

As well as ensuring that interventions are inclusive of non-pregnant women and girls, the literature on food fortification programmes and nutrition counselling stressed the importance of ensuring equity of access across different groups of women [15,18,27,74]. Socioeconomic status, educational level, ethnicity, and urban/rural status are all key factors that need to be considered if access to women and girls’ nutrition services is to be universal. Evidence from South Asia showed that GWRA from wealthier households or those utilising private health facilities were more likely to receive nutrition counselling [74]. Out-of-school adolescent girls were especially difficult to reach as schools were commonly utilised as a delivery platform for nutrition services, especially for IFA supplementation programmes [16]. Solutions may lie in the use of innovative platforms, especially for education and counselling interventions, such as social media, mHealth, behaviour change communication messaging and large-scale communication networks [86,88]. These factors and innovations should also be considered when aiming to integrate nutrition services within health systems.

Another important consideration for integrated nutrition services is consideration of programme quality. This was discussed particularly in relation to nutrition education and counselling programmes delivered by CHWs, and to CHWs working with ANC services [6,11,19,20,30]. As Berti et al. [44] described in their 2018 paper, the integration of nutrition education and counselling is most effective when messages are standardised, clear, context-specific, and supported by up-to-date guidelines. Adequate training, supervision, and monitoring systems for CHWs were also deemed to be important influences on the quality of integrated, CHW-delivered programmes [11,23,74]. Other research has indicated that the success and sustainability of integration tends to hinge on a broad whole-systems approach [84] and the careful planning of programmes to consider context-specific factors [69]. More generally, the wide scale delivery of quality nutrition programmes requires a strong health system with which to integrate into. Strengthening the building blocks of health systems according to the WHO thematic pillars [89] is therefore vitally important for successful integration of existing interventions and delivery platforms. If health systems are weak, then no matter how much effort is placed into integration, interventions will suffer from the same issues that the general health system is experiencing. To focus on strengthening a particular intervention in the absence of health system strengthening may risk exacerbating the effects of disjointed (i.e., vertical, rather than horizontal) programming.

A number of other factors were found to affect the level integration of nutrition services for GWRA. These include levels of coordination across Ministry of Health directorates and the strength of national policies and political interest. The less political interest, comprehensive policies and coordination of services, the less integrated nutrition services were found to be [69,77]. Across the literature there was a common finding that IFA supplementation for women was supported by strong policies in many countries, however, inadequate funding meant that scale-up and integration of these services was poor [12,17,62]. Furthermore, lack of awareness surrounding the importance of nutrition during pregnancy was observed as a limiting factor when identifying how well-integrated programmes were [12]. Information campaigns could help increase awareness and therefore demand for nutrition services. They could also improve adherence to micronutrient supplementation during pregnancy, another issue beyond access and integration [41,43]. Despite relatively strong integration of nutrition services with ANC services, there was a lack of convergence with health and nutrition policy at state level in this area [58].

This review also found some gaps where women and girls’ nutrition services were not being integrated into key delivery platforms. No articles were found on the integration of nutrition into postnatal care services, for example. While the WHO postnatal care (PNC) guidelines [81] include the provision of IFA supplementation for at least three months following delivery, as well as the promotion of nutrition counselling for women and their families [81], there was little evidence of this in practice across LMICs [90,91]. We found some evidence of nutrition programme integration into reproductive health services, family planning services, and adolescent health services, but evidence was limited and suggested that programmes were only delivered on a small scale, in one community/area rather than state or nationwide [18,36,71,72,76]. Lastly, while schools are not a health service, they have been recognised as a useful and cost-effective platform for delivering health interventions. However, they are under-utilised when it comes to delivering reproductive and adolescent health messages [92].

## 5. Limitations

Since our search strategy only encompassed peer-reviewed articles we did not therefore capture the grey literature detailing policy and programme integration of nutrition services for GWRA. A systematic review of country policies and their level of integration should be considered for future research. In addition, our review did not include a risk of bias assessment. However, since the focus was on lessons learned about integration, rather than intervention outcomes, this should not have had a substantial impact on our findings. Since we only discussed barriers specified by study authors, our overall assessment may be incomplete and some important barriers may have been missed, such as existence of policies, availability of training materials, incentives, etc. However, we deliberately did not conduct a full investigation into barriers as this was not a primary outcome. We specifically chose to include studies published from 2011 onwards so that the assessed literature and our systematic review was relevant to current health systems and nutrition interventions. However, restricting our time bracket may have limited our learning from older literature that could still have been valuable.

## 6. Conclusions

Access to nutrition services for women and girls is essential for the health of the current population and that of future generations. Although our review found evidence that a number of services for GWRA are well integrated into public health systems (such as ANC services, nutrition education and counselling and micronutrient supplementation programmes), the research highlighted a substantial lack of evidence surrounding integration of other nutrition services. We also identified a need for services targeting non-pregnant GWRA and postpartum women who are often overlooked. Investing in more research regarding the integration of postnatal services, family planning, home food distribution, adolescent health and reproductive health services in particular, as well as conducting an extensive search into country policies and grey literature, would provide a more comprehensive view of what is happening globally. Health system strengthening using the building blocks according to the WHO pillars [89] is a key consideration for the success of integration efforts, including those for GWRA.

## Figures and Tables

**Figure 1 nutrients-14-04488-f001:**
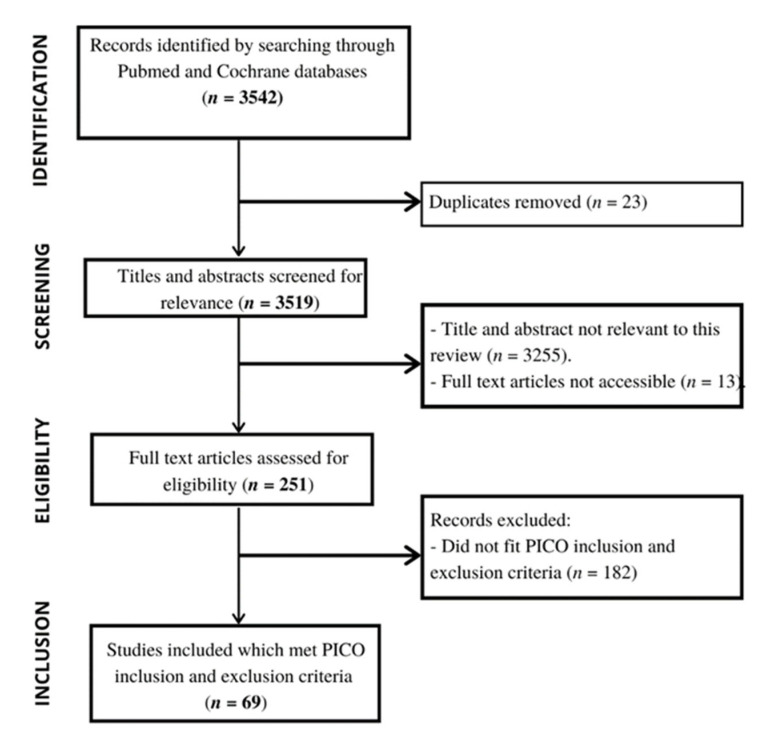
Flow diagram of the article screening process.

**Table 1 nutrients-14-04488-t001:** Summary of literature findings by nutrition intervention *.

Intervention Type	Number of Studies, Author, and Date	Public Health Systems/Policies into Which the Intervention Has Been Integrated	Key Recommendations Presented in the Study	Barriers to Integration
Food fortification	*n* = 5 Barker et al. 2018. [11] Mason et al. 2012. [12] Mgamb et al. 2017. [13] Nguyen et al. 2020. [14] Victora et al. 2012. [15]	National policy & guidelines, Integrated Child Development Services, Targeted Public Distribution System, ANC services	Update national databases to facilitate cross-country data comparison; strengthen data around maternal nutrition outcomes; further integrate into ANC services; implement pre-conception interventions; improve awareness of women’s health and nutrition with information campaigns	Weak advocacy for nutrition amongst governmental actors, lack of awareness around the importance of women’s nutrition, poor data collection and M&E systems
Home Food distribution	*n* = 1 Chakrabarti et al. 2019. [16]	National policy & guidelines, Integrated Child Development Services,	Target GWRA from low educational backgrounds	Lack of attention to equity of intervention coverage
Energy and protein supplementation	*n* = 3 De Silva et al. 2019. [4] Lassi et al. 2013. [17] Noznesky et al. 2012. [18]	National Policies, ANC services	Expand nutrition policies; improve M&E systems; implement targets interventions; invest in research on maternal and child health outcomes	Poor data collection and M&E systems; lack of awareness around the importance of women’s and adolescent nutrition; lack of research on health outcomes
Nutrition education and counselling	*n* = 25 Ajayi et al. 2013. [19] Altobelli et al. 2017. [20] Barker et al. 2019. [11] Bucher et al. 2015. [21] Chakrabarti et al. 2019. [16] De Silva et al. 2019. [4] Ghosh-Jerath et al. 2015. [22] Izudi et al. 2017. [23] Kavle et al. 2017. [24] Levin et al. 2019. [25] Mason et al. 2012. [12] Muehlhoff et al. 2017. [26] Nguyen et al. 2017. [27] Noznesky et al. 2012. [18] Riang’a et al. 2020. [28] Robert et al. 2017. [29] Ruton et al. 2018. [30] Salam et al. 2016. [31] Saldanha et al. 2012. [32] Saronga et al. 2019. [33] Sethi et al. 2019. [34] Stansert Katzen et al. 2020. [35] Varghese et al. 2014. [36] Victora et al. 2012. [15] Young et al. 2018. [37]	ANC services, Integrated Child Development Services, National policy, Population and Health Integrated Assistance Programme, Maternal, Neonatal and Child Health Programme, School Anaemia Control Programme, Primary Healthcare System, Maternal and Child Survival Programme, Baby-friendly Hospital Initiative	Target GWRA from low educational backgrounds; implement large-scale nutrition education via social media; improve awareness and demand for services via information campaigns; improve M&E systems and data collection; invest in training for CHWs; expand policies and guidelines to support CHWs; integrate nutrition counselling into family planning services; strengthen policies and programmes to postpone teenage pregnancy	Lack of attention to equity of intervention coverage, lack of prioritisation of funding for women’s health, poor data collection and M&E systems, lack of training for frontline health workers, lack of support and poor mechanisms for large-scale supervision of CHWs, lack of coordination amongst Ministry of Health directorates
Micronutrient supplementation	*n* = 47 Abdullahi et al. 2014. [38] Appiah et al. 2016. [39] Arega Sadore et al. 2015. [40] Assefa et al. 2019. [41] Babughirana et al. 2020. [42] Bannink et al. 2015. [43] Berti et al. 2018. [44] Birhanu et al. 2018. [45] Chikakuda et al. 2018. [46] De Silva et al. 2019. [4] Desta et al. 2019. [47] Digssie Gebremariam et al. 2019. [48] Dubik et al. 2019. [49] Ejigu et al. 2013. [50] Feldhaus et al. 2016. [51] Gebreamlak et al. 2017. [52] Gebremichael et al. 2019. [53] Gebremichael et al. 2020. [54] Gilder et al. 2019. [55] Jaiswal et al. 2015. [56] Kamau et al. 2020. [57] Kim et al. 2017. [58] Kiwanuka et al. 2017. [59] Lyngdoh et al. 2018. [60] Mason et al. 2012. [12] Mistry et al. 2018. [61] Nguyen et al. 2019. [62] Nguyen et al. 2017. [27] Nguyen et al. 2020. [14] Nguyen et al. 2021. [63] Omotayo et al. 2018. [64] Ouedraogo et al. 2019. [65] Paudyal et al. 2021. [66] Phillips et al. 2017. [67] Riang’a et al. 2020. [28] Roche et al. 2018. [68] Salam et al. 2016. [69] Saldanha et al. 2012. [32] Sedlander et al. 2020. [70] Sethi et al. 2019. [34] Soekarjo et al. 2018. [71] Tappis et al. 2020. [72] Thapa et al. 2016. [73] Torlesse et al. 2021. [74] Varghese et al. 2014. [36] Varghese et al. 2019. [75] Wadhwa et al. 2018. [76]	National policy, Integrated Child Development Services, Maternal, Neonatal and Child Health Services, School Anaemia Control Programme, ANC services, Primary Healthcare System, Maternal and Child Survival Programme, Reproductive Health services, Adolescent-friendly Health Clinics	Target GWRA from low educational backgrounds; invest more resources into improving coverage and accessibility; mobilise CHWs to increase awareness of the importance of supplementation; integrate health education into supplementation programmes; improve supply of tablets to ANC clinics; improve M&E systems and data collection; implement a comprehensive package delivery of Reproductive, Maternal, Newborn and Child Health Services; target non-pregnant, non-lactating adolescents or GWRA; utilise schools as a delivery platform; implement information campaigns to raise awareness and create demand for services;	Lack of attention to equity of intervention coverage, lack of awareness around the importance of women’s nutrition, poor data collection and M&E systems, lack of support and training for frontline health workers, lack of coordination amongst Ministry of Health directorates, lack of awareness around the importance of adolescent health
Total included studies:	69 (Note the unique number of studies included in this review is *n* = 69, however, many studies considered more than one type of intervention and therefore are repeated in more than one intervention row above)			

* Full details for each study are provided in Appendix A. Abbreviations: ANC—Antenatal care; CHW—Community health workers; GWRA—Girls and women of reproductive age; M&E—Monitoring and evaluation.

## Data Availability

The data presented in this study are openly available in PubMed. The data can be found here: https://pubmed.ncbi.nlm.nih.gov (accessed on 20 May 2021).

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
