# Peer review of "Integrating Women and Girls’ Nutrition Services into Health Systems in Low- and Middle-Income Countries: A Systematic Review"

_nutrients, 2022, doi:10.3390/nu14214488_

Round 1

Reviewer 1 Report

This is a well drafted scoping review on the integration of nutrition services for women and girls’ in het health system

The text is generally well written and clear.

First, my main concern is how authors defined and assessed “Barriers to integration”. Barriers can take many forms and might be difficult to identify in the papers. How did you approach this and ensure quality and consistency in the review of papers data extraction process? This needs to be better described and discussed.

Second, the approach to identify the barriers may also have led to some blind spots and incomplete assessment. There might be relevant conceptual frameworks or narratives to assist in the identification of what are barriers. Barriers such as training of health staff on nutrition, availability of policies, availability of training materials, incentives, etc; might be important as well, but, as not being mentioned explicitly in the papers, not considered in this review.

The entire section on the barriers is rather difficult to read.

A word on the limitation of the time bracket (start year of the published literature) and the databases used would be appropriate. How can this has affected findings?

My remaining comments and questions are rather minor

line 114 “and then cross-checked with the PRISMA” PRISMA is not a guideline on how to do a systematic review, only a reporting guideline.

Line 119 “we opted not to do a risk of bias since we did not want to exclude articles based on their quality”. RoB does not involve an exclusion of papers based on quality. It only assesses the risk of basis of het evidence based on the quality of the research found.

The registered protocol mentions that grey literature would be considered, but this has not happened. This needs to be clearly described and discussed if considered a source of bias.

Line 157 write “sub-Saharan” instead of “Sub-Saharan”

Author Response

Thank you to the editors and reviewers for taking time to consider and review our manuscript. We greatly appreciate the feedback, and the specific comments from the reviewers have helped us to clarify our narrative. We have addressed the comments as appropriate in the manuscript, and our point-by-point responses are provided below.

Response to reviewers

  1. First, my main concern is how authors defined and assessed “Barriers to integration”. Barriers can take many forms and might be difficult to identify in the papers. How did you approach this and ensure quality and consistency in the review of papers data extraction process? This needs to be better described and discussed. Second, the approach to identify the barriers may also have led to some blind spots and incomplete assessment. There might be relevant conceptual frameworks or narratives to assist in the identification of what are barriers. Barriers such as training of health staff on nutrition, availability of policies, availability of training materials, incentives, etc; might be important as well, but, as not being mentioned explicitly in the papers, not considered in this review. The entire section on the barriers is rather difficult to read.

Thank you for these comments. We have added more details on barriers to our outcomes section and data synthesis section. As barriers to integration were not primary outcomes, we were guided by the authors’ specification of barriers and thus did not conduct an exhaustive exploration. The above-mentioned examples in the reviewer’s comments are important ones. We have therefore restructured our section on barriers and added a sentence to our limitations section highlighting that we only captured barriers as described by authors, and may have missed other important considerations that were not explicitly dealt with in the articles. These sentences read:

Line 101: As a secondary objective, we synthesised any barriers to integration, including any factors identified by authors that restricted integration of nutrition interventions into health systems (such as lack of adequate training for community healthcare workers, poor monitoring and evaluation systems, etc.)

Line 117: Any barriers to integration specified by authors were discussed as a secondary outcomes. Outcomes and barriers were reported as they were presented by the authors.”

Line 192: “Barriers to integration fell under five main themes: lack of training and capacity building, poor multisectoral linkages and coordination, weak advocacy, lack of M&E systems, and inequity. Lack of training and support for CHWs, as well as inadequate supervision mechanisms at scale were identified as barriers by several authors [50, 63, 23, 36, 11] Lack of coordination amongst Ministry of Health directorates limited integration across health systems [50, 63], with poor linkages and coordination between programmes and policies, as well as a general lack of health and nutrition policies, acting as a barrier to integration of ANC services and micronutrient supplementation in particular [37]. The unification of multi-sectoral programmes and policies within a country is important in the successful scale-up of essential nutrition services for GWRA. However, Authors further highlighted that weak advocacy for nutrition amongst governmental actors and a lack of awareness around the importance of women and girls’ nutrition was a barrier, and meant that the most vulnerable GWRA either did not have access to, or chose not to utilise, nutrition services [45, 65].

Insufficient M&E systems were a further  barrier to integration of nutrition services for GWRA. Lack of efficient monitoring and evaluation of nutrition interventions has restricted abilities to make evidence-based adaptations or improvements to programmes, with negative impacts on the quality of services [74]. Prioritising the updating of information regarding nutrition strategies and indicators in national databases to facilitate data validation and comparison would help inform well-designed nutrition policies and programmes [53]. A final barrier commonly discussed was a lack of attention to equity of intervention coverage, leading to inequity in access to services, as many marginalised GWRA were not catered for [79, 74, 44, 69].

Line 324: “Since we only discussed barriers specified by study authors, our overall assessment may be incomplete and some important barriers may have been missed, such as existence of policies, availability of training materials, incentives, etc. However, we deliberately did not conduct a full investigation into barriers as this was not a primary outcome.”

  1. A word on the limitation of the time bracket (start year of the published literature) and the databases used would be appropriate. How can this has affected findings?

We deliberately chose this time bracket (i.e. literature from the last decade) so that our review would be relevant to current health systems. It was also a pragmatic decision to enable us to adequately handle the number of search returns with the resources we had available. However, we understand that this may have restricted our learning from previous/ older health systems/service models and have now included the following point in our limitations section:

Line 327:  “We specifically chose to include studies published from 2011 onwards so that the assessed literature and our systematic review was relevant to current health systems and nutrition interventions. However, restricting our time bracket may have limited our learning from older literature that could still have been valuable.”

  1. line 114 “and then cross-checked with the PRISMA” PRISMA is not a guideline on how to do a systematic review, only a reporting guideline.

Thank you, we have changed the wording to clarify this:

Line 114: “This review was conducted according to Synthesis Without Meta analysis (SWiM) guidelines for systematic reviews [89] and PRISMA guidelines were used as a checklist to guide our reporting (see Supplementary Table 5).”

  1. Line 119 “we opted not to do a risk of bias since we did not want to exclude articles based on their quality”. RoB does not involve an exclusion of papers based on quality. It only assesses the risk of basis of het evidence based on the quality of the research found.

Thank you, this was certainly an oversight on our part and the reviewer is correct. We have adjusted the wording regarding this in the data synthesis and limitations section. We have also requested for the PROSPERO protocol to be amended to reflect this. The revised text reads:

Line 120: “Due to the expected heterogeneity of eligible literature and outcomes, we did not perform a formal risk of bias assessment for each included piece of literature.”

Line 321: In addition, our review did not include a risk of bias assessment. However, since the focus was on lessons learned about integration, rather than intervention outcomes, this should not have had a substantial impact on our findings.”

  1. The registered protocol mentions that grey literature would be considered, but this has not happened. This needs to be clearly described and discussed if considered a source of bias.

Thank you for highlighting this. We decided not to include grey literature in our final review and so we have requested for the PROSPERO protocol to be amended to reflect this. Although it was indeed part of our original plan, we had not anticipated there being so much peer-reviewed literature available on this topic. Although this finding was extremely encouraging, it meant we needed to focus on peer-reviewed articles considering the resources available to perform this review.

  1. Line 157 write “sub-Saharan” instead of “Sub-Saharan”

Thank you, this has been changed. 

Reviewer 2 Report

This systematic review manuscript by Menezes et al. explored the existing research on the integration of nutrition services for girls and women of reproductive age (GWRA) and identified the barriers to integration. The authors conducted literature search in PubMed and Cochrane Database and identified 68 articles published since 2011. The research topic is one of the key issues in the developing and under-developed countries. The authors identified several barriers to integration and also presented key recommendations from the published articles. This is a convoluted topic as it covers a wide spectrum of population with diverse economic and cultural background.

The following are some of the comments:

The table 1 and supplement table 4 are both look similar but with some differences. This is a bit confusing. The authors claimed that the supplementary table 4 is summarized by the delivery platform and yet it is unclear.

The sum of articles in table 1 is 69 but given as 68.

L154: three studies from Asia and Pacific were mentioned but only two were cited and of which one is from Ghana.

L161: 18 articles were mentioned as focused on nutrition education and counselling, but it was 12 in table 1 and 4 in supplementary table 4. The authors could have pointed out which one they are referring.

It would be helpful for the readers to navigate through the article if the authors referred the table sections in the results.

Author Response

Reviewer 2:

  1. The table 1 and supplement table 4 are both look similar but with some differences. This is a bit confusing. The authors claimed that the supplementary table 4 is summarized by the delivery platform and yet it is unclear.

Table 1 is a summary of findings by nutrition intervention alone, whereas supplementary table 4 is summary by nutrition intervention and by delivery platform. We understand that this may be confusing to the reader, and agree this is duplicative, so we have now changed supplementary table 4 so that it is summarised by delivery platform alone. We have retained table 1 in the main text to consider the summary by nutrition intervention alone.

  1. The sum of articles in table 1 is 69 but given as 68.

Thank you for highlighting this error, this has been corrected in the table and in the flow diagram.

  1. L154: three studies from Asia and Pacific were mentioned but only two were cited and of which one is from Ghana.

Thank you for highlighting this, this has been corrected now in our text.

  1. L161: 18 articles were mentioned as focused on nutrition education and counselling, but it was 12 in table 1 and 4 in supplementary table 4. The authors could have pointed out which one they are referring. It would be helpful for the readers to navigate through the article if the authors referred the table sections in the results.

Thank you for highlighting this. Originally, we had listed all articles describing multiple interventions separately in the final row of the table. However, we understand this was confusing, and so now have changed the table to show more clearly that although there were 69 unique articles, many of them appear more than once in the table as they covered multiple interventions.

Reviewer 3 Report

Thank you for providing this review opportunity. This article examines papers and pertinent articles to discuss the nutrition of women and girls. I believe the authors used a well-organized technique and analytic advances, however, the data and its interpretation were insufficient to address the major theme of women's and girls' nutrition services preference. It is particularly difficult to finish healthcare services for women and girls in low- and middle-income nations. I would want the writers to broaden the breadth of the data sets as well as the definition of low and middle-income societies.

Author Response

Reviewer 3:

  1. Thank you for providing this review opportunity. This article examines papers and pertinent articles to discuss the nutrition of women and girls. I believe the authors used a well-organized technique and analytic advances, however, the data and its interpretation were insufficient to address the major theme of women's and girls' nutrition services preference. It is particularly difficult to finish healthcare services for women and girls in low- and middle-income nations. I would want the writers to broaden the breadth of the data sets as well as the definition of low and middle-income societies.

Thank you for your comments. We have now comprehensively addressed comments from Reviewer 1 and Reviewer 2 which we hope clarifies our narrative. We conducted our review as per our published protocol and the low- and middle-income countries included in this paper were defined by the classified World Bank list of LMICs (the countries included are also clearly listed in our search strategy as part of our supplementary materials). Our data was informed by our search strategy and increasing the breadth of our data sets would have been outside the scope of this paper, although we agree that ideally with more time and resources we could have captured much more. However, we trust what we have provided is still a useful starting point for discussion from which others can build.  

Round 2

Reviewer 3 Report

Dear Writers,

Thank you for your thorough revisions.

As guided in the first round of assessment, it seemed appropriate.

I would recommend that this manuscript be published.